

# RA-QoS: a robust autoencoder-based QoS predictor for highly accurate web service QoS prediction

Shun Fu, Junnan Li and Lufeng Wang

Chongqing Industry Polytechnic College, Chongqing, China

## ABSTRACT

Web services are fundamental for online service-oriented applications, where accurately predicting quality of service (QoS) is critical for recommending optimal services among multiple candidates. Since QoS data often contains noise—stemming from factors like remote user or service locations—current deep neural network (DNN)-based QoS predictors, which generally rely on L2-norm loss functions, face limitations in robustness due to sensitivity to outliers. To address this issue, we propose a novel robust autoencoder-based QoS predictor (RA-QoS) that leverages a hybrid loss function combining bias, training bias, L1-norm and L2-norm to build a robust Autoencoder. This hybrid approach allows RA-QoS to better handle noisy data, minimizing the impact of outliers and biases on prediction accuracy. The RA-QoS model further incorporates preprocessing and training biases, improving its adaptability to real-world QoS data. To evaluate the proposed RA-QoS predictor, extensive experiments are conducted on two real-world QoS datasets. The results demonstrate that our RA-QoS predictor exhibits superior robustness to outliers and higher accuracy in QoS prediction compared to the related state-of-the-art models.

## INTRODUCTION

With the growing number of web services such as online video platforms, online meeting apps, social media websites, and cloud computing applications, more and more web services are providing similar functionality. Moreover, with the vastness of the world and widespread internet adoption, web services may run on different client devices and operating systems. Thus, selecting the most suitable services from numerous candidates for clients is crucial in practical web services applications (*Zheng et al., 2020*; *Jia et al., 2022*).

Quality of service (QoS) is typically used to describe the non-functional attribute features of services, including response time, throughput, *etc.* (*Sathya et al., 2010*). If all the QoS data are obtained, service selection can be easy to implement. However, it is not realistic and cost-effective to collect the QoS data for all services. Typically, QoS data are assessed from the perspective of each user and vary from person to person. Each user usually only invokes a small set of web services, implying that the majority of web services remain uncalled. Consequently, the QoS data for most services are unknown. Therefore,

Corresponding author
Shun Fu, fushun@cqipc.edu.cn

the problem of QoS Prediction becomes the primary challenge in web services applications.

Problem definition: Given a set of users $\mathcal{U} = u_1, u_2, \ldots, u_m$ and a set of services $\mathcal{S} = s_1, s_2, \ldots, s_n$, where each user $u_i$ can invoke certain services $s_j$ and obtain corresponding QoS data, the known QoS data can be represented as a sparse matrix $Q \in \mathbb{R}^{m \times n}$, where $q_{ij}$ denotes the QoS measurement (*e.g.*, response time, throughput) of user $u_i$ for service $s_j$. The objective of the problem is to predict the unknown values in this matrix, *i.e.*, estimate $q_{ij}$ where $q_{ij}$ is missing in the original matrix.

Collaborative filtering (CF) is one of the most successful personalized prediction techniques for recommender systems. Owing to its simple and easy-to-understand, strong interpretability, and other advantages, it has been widely applied to QoS prediction in web service (*Zheng et al., 2020*; *Wu et al., 2017*; *Botangen et al., 2020*). However, CF-based QoS prediction methods also have the shortcoming that they cannot capture the high-dimensional and nonlinear complex relationship between users and services. Therefore, researchers begin to apply deep neural networks (DNNs) to QoS data prediction (*Liang et al., 2021*; *Yin et al., 2020*; *Ma, Geng & Wang, 2020*).

QoS data typically gathered from a multitude of users across diverse scenarios, often exhibit pervasive biases and outliers due to many reasons (*Yang et al., 2018a*). For instance, some users/services are located in remote areas. Their response time of QoS is much larger than the closer users/services. Another example is that biases originating from the service side are also prevalent in collected QoS data because popular services tend to garner more user attention, resulting in crowded user behaviors towards services. Therefore, biases and outliers are commonly mixed with QoS data, which poses significant challenges for DNNs-based QoS data prediction (*Ma et al., 2020*; *Luo, Wang & Shang, 2019*).

Recently, although application of DNNS-based models in QoS prediction has been extensively explored (*Liang et al., 2021*; *Yin et al., 2020*), they only consider relatively scattered biases while lacking comprehensive solutions. Moreover, their shared nature lies in the L2-norm-oriented loss function, which limits their robustness to outliers because the L2-norm is sensitive to noisy data (*Li et al., 2017*). Consequently, their ability to accurately predict QoS data diminishes significantly in the presence of substantial noise.

**Why Autoencoder for QoS prediction**: To address these limitations, we employ an autoencoder for QoS prediction due to its capacity to learn low-dimensional representations that capture underlying data structures, even in the presence of sparse and noisy data. Autoencoders excel at reconstructing missing information in incomplete datasets, making them well-suited for predicting unknown QoS values. Additionally, autoencoders offer flexibility in combining L1 and L2 norms in the loss function, providing a mechanism for robustly handling outliers and various biases. This property enables our model to maintain high prediction accuracy even when data are noisy or contain substantial biases.

**Main novelty**: Based on these advantages, this article introduces a robust autoencoder-based QoS predictor (RA-QoS) with two key innovations. Firstly, RA-QoS incorporates a robust loss function specifically designed to handle both biases and outliers

by combining preprocessing bias (PB), training bias (TB), and both L1-norm and L2-norm elements. Secondly, RA-QoS applies a dual-norm structure to improve robustness and reduce sensitivity to noise, enhancing prediction accuracy in noisy environments. Experiments on two real-world QoS datasets validate that RA-QoS outperforms related models in terms of both robustness and accuracy.

This article has the following main contributions:

(1) It proposes a novel RA-QoS predictor. The proposed RA-QoS predictor can achieve robust and accurate QoS data prediction when QoS data are mixed with much noise data.

(2) It provides detailed theoretical analyses as well as algorithm designs for the proposed RA-QoS predictor.

(3) It conducts multiple comparative experiments, including model adjustments, hyperparameter tuning, and typical ablation studies, to validate that the proposed RA-QoS predictor indeed enhances the accuracy of QoS prediction.

## RELATED WORKS

### Collaborative filtering-based QoS prediction

Collaborative filtering (CF) stands as a leading prediction method in recommendation systems and is extensively utilized in Web service QoS prediction (*Zheng et al., 2020*). CF leverages historical data on user behavior to detect similarities among users or items and forecast user preferences for items, incorporating quality assessments from known services to estimate for unseen users. Due to its effectiveness in capturing the dynamics of user and service characteristics, CF-based methods have enabled personalized QoS predictions for service users. Several influential models have emerged from this approach, including kernel least-mean-square (KLMS) by *Luo et al. (2016)*, embedding and factorization methods by *Wu et al. (2017)*, matrix factorization by *Zheng & Lyu (2013)*, location-based factorization by *Yang et al. (2018b)*, and tensor techniques within a five-dimensional QoS framework by *Wang et al. (2016)*. These models demonstrate strengths in user-service dynamics but often lack comprehensive solutions to handle complex, nonlinear relationships and varying biases.

### DNN-based QoS prediction

Deep neural networks (DNNs), with their strong nonlinear learning capabilities, have also gained traction for QoS prediction. Notable models include recurrent networks for classification and regression by *Chen et al. (2019)*, a stacked denoising autoencoder for long-tail services by *Bai et al. (2017)*, and the hybrid NRR model by *Li et al. (2017)*. More recent work includes GNN-based CF models by *Zhang & Chen (2019)* and *Ying et al. (2018)*, which exploit user-item graph preferences. While these methods improve QoS prediction accuracy, they fall short in addressing biases and outliers in QoS data comprehensively.

DNN-based models incorporating partial debiasing (*e.g.*, *Schnabel et al., 2016*; *Saito, 2020*; *Yuan, Luo & Shang, 2018*) or combining L1 and L2 norms (*e.g.*, *Zhu et al., 2018*; *Raza & Ding, 2020*; *Wu et al., 2023*; *Huang et al., 2023*) have been introduced, yet they lack robustness to diverse biases and outliers. Existing L1 and L2 norm-based methods address

**Table 1 Comparison of related QoS prediction methods.**

| Method | Merits | Demerits |
|---|---|---|
| Collaborative filtering (CF) (*Zheng et al., 2020*) | Simple, interpretable, effective for capturing user-service similarities | Struggles with high-dimensional, nonlinear relationships; partial bias consideration |
| Kernel least-mean-square (KLMS) (*Luo et al., 2016*) | Captures QoS relationships effectively | Limited in robustness to outliers |
| Embedding + Factorization (*Wu et al., 2017*) | Increases prediction accuracy using latent factors | High sensitivity to noise due to L2-norm loss |
| DNN-based models (*e.g., Chen et al., 2019; Bai et al., 2017; Kim et al., 2024*) | Strong nonlinear modeling capacity, suitable for sparse data | Lacks comprehensive bias handling; sensitive to outliers due to L2-norm reliance |
| Graph neural networks (GNNs) (*Zhang & Chen, 2019; Ying et al., 2018; Kim et al., 2024*) | Utilizes user-item graph preferences, effective for user-item similarity | Limited in handling diverse biases; does not address outlier issues comprehensively |
| Position-aware low-rank matrix factorization (LLMF) (*Zhu et al., 2018*) | Robust to certain biases using L1-norm | Instability and non-uniqueness issues with L1-norm; limited scalability |
| Proposed RA-QoS | Comprehensive bias handling, robust to outliers with combined L1/L2 loss function | To be validated further in real-time applications |

specific biases but struggle with the instability and non-uniqueness issues associated with the L1 norm.

In response to these limitations, this study proposes a novel RA-QoS predictor with a robust loss function combining preprocessing bias (PB), training bias (TB), and both L1 and L2 norms within an autoencoder framework. This approach enables comprehensive handling of biases and improved robustness against outliers, thereby enhancing prediction accuracy.

The comparison of related QoS prediction methods are summerized in Table 1.

## PRELIMINARIES AND PROBLEM DEFINITIONS

**User-service interaction matrix**. Given a user set $U = \{u_1, u_2, \ldots, u_m\}$ and service set $S = \{s_1, s_2, \ldots, s_n\}$, the matrix $Q \in \mathbb{R}^{m \times n}$ with $m$ rows and $n$ columns denotes the relationship between $U$ and $S$. Each element in matrix $Q$, *i.e.*, $q_{ij}$, denotes the element at $i$-th row and $j$-th column in matrix $Q$. The $q_{ij}$ is the QoS value of user $i$ calling service $j$. For example, the $q_{12}$ could be the response time for service $s_2$ on user $u_1$. As shown in Fig. 1, the user-service interaction matrix consists of rows of users and columns of services. In real-world data, a significant portion of the elements in the user-service matrix are blank. This is because it is impractical to observe all service quality assessments for all the users. Such that we define a binary matrix $D^{|U| \times |S|} \in \{0, 1\}$ differentiates between observed and unobserved interactions in $Q$. For each element $d_{mn}$ in $D$, we have:

$$d_{mn} = \begin{cases} 1, & \text{if } q_{mn} \text{ is observed.} \\ 0, & \text{otherwise} \end{cases} \tag{1}$$

**Problem of QoS prediction**. The QoS prediction problem is defined to make predictions for the unobserved ratings of $Q$ by learning a parametric model $f(\cdot)$ from the observed ratings of $Q$ as follows:

$$f(U, S; \theta) \mapsto Q \tag{2}$$

|      | s1   | s2   | s3   | ... |
|------|------|------|------|-----|
| u1   | 1.72 | 0.56 |      |     |
| u2   |      | 1.87 |      |     |
| u3   | 0.81 |      | 3.14 |     |

...

**Figure 1  User-service interaction matrix which displays the quality ratings given by users to various services.** Specifically, user u1 rated service s1 with a score of 1.72 and service s2 with a score of 0.56; user u2 rated service s1 with a score of 1.87; and user u3 rated service s1 with a score of 0.81 and service s3 with a score of 3.14. Empty element of that matrix indicate missing data.

where $\theta$ indicates the parameters of $f(\cdot)$. The objective function $f(\cdot)$ indicates to minimize the *Empirical Risk* as follows:

$$L(f) = \sum_{u_i \in U, s_j \in S} \varepsilon(f(u_i, s_j; \theta), q_{ij}). \tag{3}$$

## THE PROPOSED METHOD: RA-QoS MODEL

### Autoencoder input and output

The RA-QoS model (shown in Fig. 2) utilizes an autoencoder architecture designed to predict missing QoS values. The input to the autoencoder, denoted as $y^{(q)}$, represents observed QoS data from the user-service matrix, specifically response time or throughput values for a given user-service pair. This input data may be sparse due to the incomplete nature of user-service interactions.

The autoencoder's objective is to learn a compressed representation of the input data through its hidden layers, which is then used to reconstruct or predict the output. The output of the autoencoder, denoted as $\hat{y}^{(q)}$, is the predicted QoS value corresponding to the input. The model is trained to minimize the difference between the actual QoS value (from the observed data) and the predicted value $\hat{y}^{(q)}$, allowing it to predict missing values effectively.

The architecture of the RA-QoS model is shown in Fig. 3. In the model, the input layer is on the left, denoted by $y^{(q)}$. The input data then passes through a series of hidden layers $(l_1, l_2, \ldots, l_k)$, where it undergoes transformations by applying a series of weights $(w_k)$ and biases $(b_k)$, as well as nonlinear activation function such as Sigmoid function. The result of

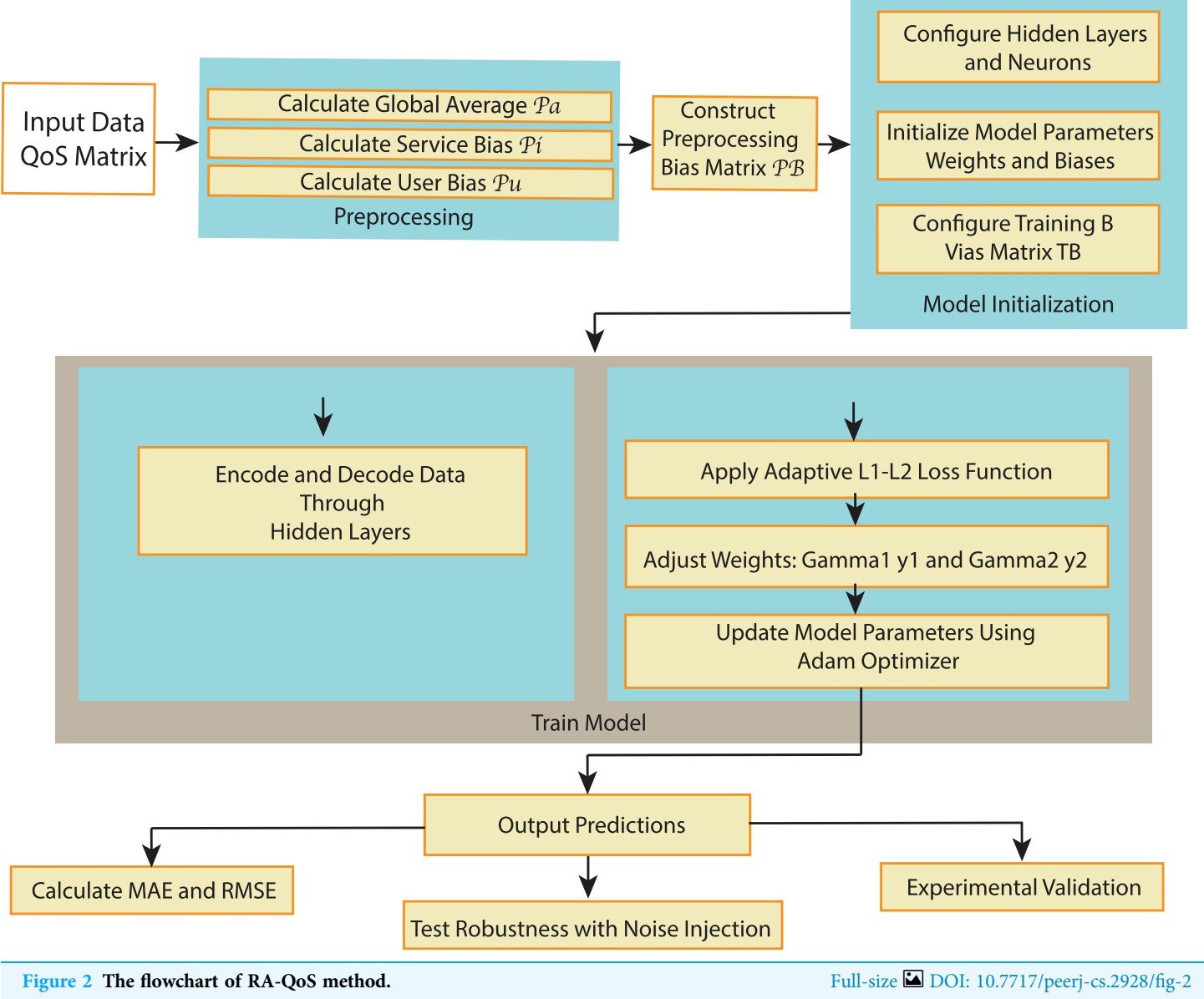

**Figure 2** The flowchart of RA-QoS method.

each transformation is passed to the next layer until reaching the final layer ($l_k$). From there, outputs are generated to make predictions, denoted as $\hat{y}^{(q)}$. Note the parameters in the $\phi(TB)$ are trainable. We adopt the $\phi(TB)$ and $\psi(PB)$ to represent TB and PB combinations.

## Calculation of PB and TB

The preprocessing bias (PB) combinations denote to preprocess the observed user behavior data before training. They comprises three components: $Pa$, $Pi^{(n)}$ and $Pu^{(m)}$, representing the global average rating, service bias, and user bias, respectively. These combinations are varied using $z_1$, $z_2$ and $z_3$ to adjust $Pa$, $Pi^{(n)}$ and $Pu^{(m)}$ independently.

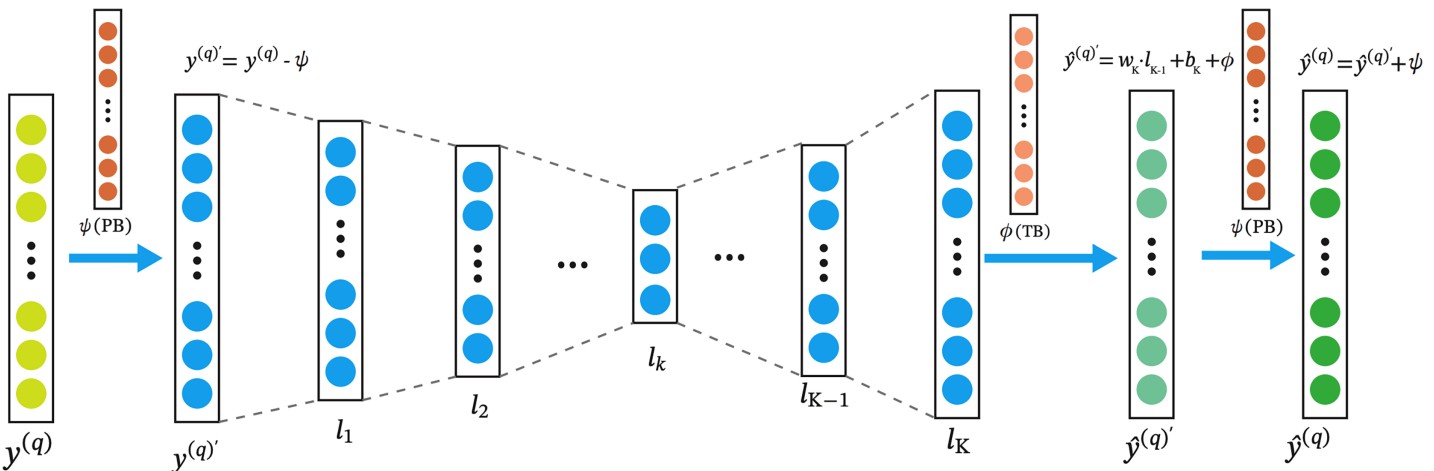

**Figure 3 The architecture of RA-QoS model.** The y(q) on the left is the the input layer. The input data then passes through a series of hidden layers $(l_1, l_2, \ldots, l_k)$, where it undergoes transformations by applying a series of weights (wk) and biases (bk), as well as nonlinear activation function such as Sigmoid function. The result of each transformation is passed to the next layer until reaching the final layer (lk). From there, outputs are generated to make predictions, denoted as $\hat{y}^{(q)}$.

The global average rating $Pa \in \mathbb{N}^+$ reflects the statistical properties of observed QoS ratings in the user-service matrix $Q$. It can be obtained as follows:

$$Pa = z_1 \cdot \frac{\sum_{mn \in \Gamma} q_{mn}}{\sum_{mn \in \Gamma} d_{mn}}, \tag{4}$$

where $\Gamma \in \mathbb{R}^{|\mathbb{M}| \times |\mathbb{N}|}$ represents the training set that is a subset of the matrix $Q$. The preference of all users for a service $s$ can be estimated using $Pi^{(n)}$ with the equation as follows:

$$Pi^{(n)} = z_2 \cdot \frac{\sum_{m \in \Gamma(n)} d_{mn} \cdot (q_{mn} - Pa)}{\omega_1 + \sum_{m \in \Gamma(n)} d_{mn}}, \tag{5}$$

where $\Gamma(n)$ represents the $n$-th column of $\Gamma$, and $\omega_1$ denotes the threshold constant associated with the average rating of service $s$. The preference of a user for different services $Pu$ can be noted as:

$$Pu^{(m)} = z_3 \cdot \frac{\sum_{n \in \Gamma(m)} d_{mn} \cdot \left(q_{mn} - Pa - Pi^{(n)}\right)}{\omega_2 + \sum_{n \in \Gamma(m)} d_{mn}}. \tag{6}$$

Table 2 summarizes all the possible combinations of PB. Moreover, $Pi$ and $Pu$ denote the PB vectors for all services and all users, respectively.

The training bias (TB) operates in conjunction with the training process, unlike PB. TB consists of two components: $Ti(s)$ and $Tu(u)$, which extract user $u$'s and service $s$'s preferences, respectively, during training. Similarly, we employ $z_4$ and $z_5$ to modulate the values of $Ti(s)$ and $Tu(s)$ to create various TB combinations, and $Ti$ and $Tu$ represent the TB vectors for all services and all users, respectively. A summary of all possible TB combinations is provided in Table 3.

**Table 2 All possible cobinations of PB.**

|  | PB.1 | PB.2 | PB.3 | PB.4 | PB.5 | PB.6 | PB.7 | PB.8 |
|---|---|---|---|---|---|---|---|---|
| Value of weights | $z_1 = 0$ | $z_1 = 1$ | $z_1 = 0$ | $z_1 = 1$ | $z_1 = 1$ | $z_1 = 0$ | $z_1 = 0$ | $z_1 = 1$ |
|  | $z_2 = 0$ | $z_2 = 1$ | $z_2 = 1$ | $z_2 = 0$ | $z_2 = 1$ | $z_2 = 0$ | $z_2 = 1$ | $z_2 = 0$ |
|  | $z_3 = 0$ | $z_3 = 1$ | $z_3 = 1$ | $z_3 = 1$ | $z_3 = 0$ | $z_3 = 1$ | $z_3 = 0$ | $z_3 = 0$ |

**Table 3 All possible cobinations of TB.**

|  | TB.1 | TB.2 | TB.3 | TB.4 |
|---|---|---|---|---|
| Value of weights | $z_4 = 0$ | $z_4 = 1$ | $z_4 = 1$ | $z_4 = 0$ |
|  | $z_5 = 0$ | $z_5 = 1$ | $z_5 = 0$ | $z_5 = 1$ |

## The combinations of PB and TB

The RA-QoS model adjusts the combination of PB and TB to balance the influence of bias. In this model, the $l_k$ is used to represent the $k$-th hidden layer of RA-QoS, $k \in \{1, 2, \ldots, K\}$. The term $w_k$ represents the weighted term of $l_k$, and the term $b_k$ represents the weighted term of $l_k$. Therefore, RA-QoS's prediction function can be expressed as follows:

$$
\begin{aligned}
&\mathbf{y}^{(q)'} = \mathbf{y}^{(q)} - \psi \\
&l_1 = g\left(w_1 \cdot \mathbf{y}^{(q)'} + b_1\right) \\
&l_k = g(w_k \cdot l_{k-1} + b_k), k = 2, \cdots, K-1 \\
&\widehat{\mathbf{y}}^{(q)'} = w_K \cdot l_{K-1} + b_K + \phi \\
&\widehat{\mathbf{y}}^{(q)} = f\left(\mathbf{y}^{(q)}; \theta, \phi, \psi\right) = h(l_K + \psi) \\
&\begin{cases}
\text{I-RA-QoS}: \begin{cases}
\mathbf{y}^{(q)} = \mathbf{y}^{(n)} = \left\{y_{1n}, \cdots, y_{|M|n}\right\} \\
\psi = Pa + Pi^{(n)} + Pu \\
\phi = Ti^{(n)} + Tu
\end{cases} \\
\text{U-RA-QoS}: \begin{cases}
\mathbf{y}^{(q)} = \mathbf{y}^{(m)} = \left\{y_{m1}, \cdots, y_{m|N|}\right\} \\
\psi = Pa + Pi + Pu^{(m)} \\
\phi = Ti + Tu^{(m)}
\end{cases}
\end{cases}
\end{aligned}
\tag{7}
$$

The model is optimized by minimizing the following objective function:

$$
\begin{cases}
\text{I-RA-QoS}: \begin{cases}
L(f) = \sum_{\substack{\mathbf{y}^{(n)} \in Y \\ \mathbf{d}^{(n)} = D}} \left(\left(\mathbf{y}^{(n)} - \widehat{\mathbf{y}}^{(n)}\right) \odot \mathbf{d}^{(n)}\right)^2 \\
+ \frac{\lambda_1}{2} \cdot \left(\sum_{k=1}^{K} (w_k)^2\right) + \frac{\lambda_2}{2} \cdot \left(Ti^{(n)} + Tu\right)^2
\end{cases} \\
\text{U-RA-QoS}: \begin{cases}
L(f) = \sum_{\substack{\mathbf{y}^{(m)} \in Y \\ \mathbf{d}^{(m)} = D}} \left(\left(\mathbf{y}^{(m)} - \widehat{\mathbf{y}}^{(m)}\right) \odot \mathbf{d}^{(m)}\right)^2 \\
+ \frac{\lambda_1}{2} \cdot \left(\sum_{k=1}^{K} (w_k)^2\right) + \frac{\lambda_2}{2} \cdot \left(Ti + Tu^{(m)}\right)^2
\end{cases}
\end{cases}
\tag{8}
$$

where $\lambda_1, \lambda_2$ are the regularization rate hyper-parameters. To minimize the function $L(f)$ and train the learnable parameters $\Theta = \{w_1, b_1, \ldots, w_K, d_k, Tu, Ti\}$ above, we adopted Adam (*Kingma & Ba, 2014*) to adapt the learning rate automatically to train $\Theta$.

## Self loss function in RA-QoS

The L2-norm loss function is sensitive to outliers in QoS data for each user, while the L1-norm loss function lacks stability during training. To address this, we combine them into a self-adaptively weighted L1-and-L2-norm-oriented loss function in RA-QoS model. Thus, the objective function with L1-and-L2norm-oriented loss function of RA-QoS becomes:

$$
\begin{cases}
\text{I-RA-QoS :} & \begin{cases} L(f) = \sum_{\substack{\mathbf{y}^{(n)} \in \mathrm{Y} \\ \mathbf{d}^{(n)} = D}} \gamma_1 \cdot \left| \left( \mathbf{y}^{(n)} - \widehat{\mathbf{y}}^{(n)} \right) \odot \mathbf{d}^{(n)} \right| + \gamma_2 \cdot \left( \left( \mathbf{y}^{(n)} - \widehat{\mathbf{y}}^{(n)} \right) \odot \mathbf{d}^{(n)} \right)^2 \\ + \frac{\lambda_1}{2} \cdot \left( \sum_{k=1}^{K} (w_k)^2 \right) + \frac{\lambda_2}{2} \cdot \left( Ti^{(n)} + Tu \right)^2 \end{cases} \\
\text{U-RA-QoS :} & \begin{cases} L(f) = \sum_{\substack{\mathbf{y}^{(m)} \in \mathrm{Y} \\ \mathbf{d}^{(m)} = D}} \gamma_1 \cdot \left| \left( \mathbf{y}^{(m)} - \widehat{\mathbf{y}}^{(m)} \right) \odot \mathbf{d}^{(m)} \right| + \gamma_2 \cdot \left( \left( \mathbf{y}^{(m)} - \widehat{\mathbf{y}}^{(m)} \right) \odot \mathbf{d}^{(m)} \right)^2 \\ + \frac{\lambda_1}{2} \cdot \left( \sum_{k=1}^{K} (w_k)^2 \right) + \frac{\lambda_2}{2} \cdot \left( Ti + Tu^{(m)} \right)^2 \end{cases}
\end{cases} \tag{9}
$$

where $\gamma_1$ and $\gamma_2$ denote the aggregation weights. They self-adaptively regulate the influence of L1-norm and L2-norm, respectively. Note that to maintain the magnitude of loss, $\gamma_1 + \gamma_2 = 1$, and $\gamma_1 \cdot \gamma_2 \geq 0$. The aggregation strategy has the following design strategy: increasing $\gamma_1$ if the partial loss of L1-norm is lower than L2-norm, otherwise increasing $\gamma_2$. In training process, let $\delta_1^t$ and $\delta_2^t$ denote the partial losses of $L_1$-norm, $L_2$-norm and let $\delta_{1,2}^t = \delta_1^t + \delta_2^t$. Let $A_1^t = \sum_{j=1}^t \delta_1^j$, $A_2^t = \sum_{j=1}^t \delta_2^j$ and $A_{1,2}^t = \sum_{j=1}^t \delta_{1,2}^j$.

## L1-L2 norm fusion mechanism and its contribution to robustness

The RA-QoS model incorporates a self-adaptive fusion mechanism between the L1-norm and L2-norm, which significantly contributes to its robustness in the presence of noise and irregular data. The aggregation weights $\gamma_1$ and $\gamma_2$ govern the influence of the L1-norm and L2-norm, respectively, with the constraint $\gamma_1 + \gamma_2 = 1$ and $\gamma_1 \cdot \gamma_2 \geq 0$. This mechanism allows the model to dynamically adjust its focus between the two norms based on the current training stage.

The design strategy of this fusion mechanism is as follows: when the partial loss of L1-norm $\delta_1^t$ is lower than the L2-norm loss $\delta_2^t$, the model increases $\gamma_1$, favoring the L1-norm to better handle sparse or noisy data. Conversely, if the L1-norm loss is larger, the model increases $\gamma_2$, which places more emphasis on the L2-norm. This self-adaptive strategy allows the model to dynamically adjust its sensitivity to noise and outliers throughout the training process.

At each training iteration, the partial losses of the L1-norm ($\delta_1^t$) and L2-norm ($\delta_2^t$) are computed, and their cumulative sums are updated as follows:

$$
A_1^t = \sum_{j=1}^{t} \delta_1^j, \quad A_2^t = \sum_{j=1}^{t} \delta_2^j, \quad A_{1,2}^t = \sum_{j=1}^{t} (\delta_1^j + \delta_2^j).
$$

This mechanism ensures that the model adapts its loss function dynamically during training. The aggregation of the L1-norm and L2-norm helps to balance the model's sensitivity to sparse, noisy, or outlier data points while maintaining stability in learning.

As a result, this self-adaptive fusion mechanism makes the RA-QoS model particularly robust when confronted with noisy, sparse, or imbalanced data. By giving more weight to

the L1-norm when dealing with outliers and noise, and adjusting to the L2-norm when the data is more consistent, the model improves its generalization and performance under challenging conditions. This enhanced robustness ensures that the RA-QoS model performs stably even when faced with imperfections in the data, making it an excellent choice for real-world QoS prediction tasks.

The detailed algorithm is shown in Algorithm 1.

## Time and space complexity

Providing $|M|$ inputs to the hidden units of I-RA-QoS requires $|M|$ multiplications and $|M| + 1$ additions to account for bias terms and activation functions. The computational demand of these operations is linear, represented as $O(|M|)$. If I-RA-QoS has $x$ hidden units in its first layer, each receiving $|M|$ inputs, the complexity for this layer is calculated as $x \times O(|M|)$. With two hidden layers, the total time complexity of I-RA-QoS is $x \times O(|M|) + O(|M|) \times x \times O(|M|)$, simplifying to $O(|M|^2 \times x)$. Assuming $x$ remains constant, the complexity stabilizes at $O(|M|^2)$. Similarly, the time complexity for U-RA-QoS can be deduced to be $O(|N|^2 \times x)$ and $O(|N|^2)$, using the same computational framework.

RA-QoS only needs to keep a version of the input matrix and additional record information with similar dimensions. In contrast, I-RA-QoS requires $O(|M| \times K)$ space to store its parameters. Consequently, the space complexity for I-RA-QoS is $O(|M| \times |N| + |M| + K)$. Given that $N$ typically far exceeds $K$, the space complexity for I-RA-QoS simplifies to $O(|M| \times |N|)$, a simplification that also holds true for U-RA-QoS.

## EXPERIMENTS

### Quality criteria for services

In this study, we focus on two commonly used quality of service (QoS) metrics: response time and throughput. Response time is defined as the time taken from when a user sends a request until the service responds, and it reflects the promptness of service delivery. Throughput is defined as the rate at which data is successfully processed by a service over a specific period, indicating the capacity of the service to handle user demands. Both metrics are obtained from the WS-Dream dataset (*Zheng et al., 2010*), where they were recorded by users interacting with various web services. To evaluate and compare the prediction accuracy of the proposed RA-QoS model, these metrics serve as target values in our prediction model. Errors between the predicted and actual values of these metrics are calculated using mean absolute error (MAE) and root mean squared error (RMSE), as detailed below.

### Platform and environment

The experiments were conducted on a computing platform equipped with Intel(R) Core (TM) i7-10510U CPU @ 1.80 GHz 2.30 GHz, IVIDIA RTX3090 GPU and 16.0 GB RAM. The RA-QoS model and baseline algorithms were implemented using Python and several machine learning libraries, including tensorflow for deep learning functionality. Experiments were executed with additional packages for data processing and evaluation,

---

**Algorithm 1** Training procedure for RA-QoS model.

**Input:** User-service matrix $Q \in \mathbb{R}^{|M| \times |N|}$, Preprocessing Bias (PB) combinations $\psi$, Training Bias (TB) combinations $\phi$, maximum iterations $T$, learning rate $\mu$, regularization factors $\lambda_1, \lambda_2$

**Output:** Trained model parameters $\theta$

1 Initialize model parameters $\theta = \{w_1, \ldots, w_K, b_1, \ldots, b_K\}$;

2 Preprocess user behavior data to compute PB: *Pa, Pi, Pu*;

3 Initialize accumulative losses $A_1 = 0, A_2 = 0, A_{1,2} = 0$;

4 **for** *iteration $t = 1$ **to** $T$* **do**

5      **for** *each observed entry $(m, n)$ in $Y$* **do**

6          Compute $\mathbf{y}^{(q)} = \mathbf{y}^{(n)}$ for I-RA-QoS and $\mathbf{y}^{(q)} = \mathbf{y}^{(m)}$ for U-RA-QoS;

7          Subtract PB from input data: $\mathbf{y}^{(q)'} = \mathbf{y}^{(q)} - \psi$;

8          Forward pass through the model to get intermediate layer outputs $l_1, \ldots, l_{K-1}$;

9          Compute pre-output: $\widehat{\mathbf{y}}^{(q)'} = w_K \cdot l_{K-1} + b_K + \phi$;

10         Add PB to get final output: $\widehat{\mathbf{y}}^{(q)} = \widehat{\mathbf{y}}^{(q)'} + \psi$;

11         Compute partial losses: $\delta_1^t = \gamma_1 \cdot ||(\mathbf{y}^{(q)} - \widehat{\mathbf{y}}^{(q)}) \odot \mathbf{d}^{(q)}||_1, \delta_2^t = \gamma_2 \cdot ||(\mathbf{y}^{(q)} - \widehat{\mathbf{y}}^{(q)}) \odot \mathbf{d}^{(q)}||_2^2$;

12         Update accumulative losses: $A_1 = A_1 + \delta_1^t, A_2 = A_2 + \delta_2^t, A_{1,2} = A_{1,2} + \delta_{1,2}^t$;

         /* Compute gradients */

13         Compute gradient $\nabla_\theta L$ based on the loss function;

         /* Update model parameters using Adam optimizer */

14         **for** *each parameter $\theta_i$ in $\theta$* **do**

15             $m_i = \beta_1 \cdot m_i + (1 - \beta_1) \cdot \nabla_{\theta_i} L$;

16             $v_i = \beta_2 \cdot v_i + (1 - \beta_2) \cdot (\nabla_{\theta_i} L)^2$;

17             $\hat{m}_i = m_i / (1 - \beta_1^t)$;

18             $\hat{v}_i = v_i / (1 - \beta_2^t)$;

19             $\theta_i = \theta_i - \alpha \cdot \hat{m}_i / (\sqrt{\hat{v}_i} + \varepsilon)$;

20         **end**

21      **end**

22      Update $\gamma_1$ and $\gamma_2$ based on accumulative losses.

23 **end**

---

such as NumPy. The source code for experiment is available at this URL (https://gitee.com/sonica/RA-QoS).

## Datasets

In this work, the WS-Dream dataset (*Zheng et al., 2010*) was employed for model experimentation. This dataset is a real-world web service dataset that has been frequently utilized in prior studies. The experiment utilized two benchmark QoS datasets: Response Time (RT) and Throughput (TP). The Response Time dataset contains 1,873,838 records, while the Throughput dataset contains 1,831,253 records. Both datasets were generated by 339 users across 5,825 distinct services.

For each experiment, different ratios of training data were used, as indicated by the ID in the table. Specifically, the RT in the ID column refers to the Response Time dataset, and the TP in the ID column refers to the Throughput dataset. The number following the underscore in the ID (*e.g.*, RT_05) denotes the ratio of training data used in that particular experiment set, where 05 corresponds to 5% of the dataset used for training.

Both datasets were partitioned into training and testing sets at four different ratios: 5%, 10%, 15%, and 20%. The exact number of samples used for training and testing at each ratio is provided in Table 4.

## Evaluation of experiments

The task of QoS prediction involves forecasting missing data in sparse QoS matrices. Prediction, by its nature, cannot achieve 100% accuracy; thus, there always exists an error between predicted and actual values. Therefore, we utilize this error to evaluate the accuracy of QoS prediction on the test set. MAE and RMSE are usually used as the error evaluation metrics. MAE addresses the issue of error cancellation, accurately reflecting the magnitude of prediction errors. RMSE is sensitive to larger or smaller errors in the test set, effectively reflecting the robustness of predictions. Hence, this study selects RMSE and MAE as evaluation metrics for assessing predicted QoS values. The specific formulas are as follows:

$$\text{MAE} = \frac{\sum_{u,s} \left| Q_{u,s} - \hat{Q}_{u,s} \right|}{N} \tag{10}$$

$$\text{RMSE} = \sqrt{\frac{\sum_{u,s} \left( Q_{u,s} - \hat{Q}_{u,s} \right)^2}{N}} \tag{11}$$

where $Q_{u,s}$ represents the actual values in the test set, and $\hat{Q}_{u,s}$ represents the predicted values given by the proposed RA-QoS model.

## Baselines

To validate the performance of the RA-QoS model in web service quality prediction, this study compares four mainstream QoS prediction methods:

Robust Sparse Non-negative Matrix Factorization (RSNMF) (*Peng et al., 2018*): RSNMF is utilized for the decomposition of user-item rating matrices, facilitating personalized recommendations. By learning feature representations of users and items, RSNMF captures user interests and item attributes, thus achieving accurate recommendations.

Non-negative Alternating Matrix Factorization (NAMF) (*Li et al., 2022*): NAMF is also a matrix factorization technique characterized by its alternating update optimization process.

Data-Characteristic-Aware Latent Factor (DCALF) (*Wu et al., 2022*): DCALF is a latent factor model used in recommendation systems, considering the influence of data characteristics. The DCALF model enhances latent factor models by incorporating additional data features, thus better capturing the relationships between users and items.

Autoencoders Meet Collaborative Filtering (AutoRec) (*Sedhain et al., 2015*): AutoRec is a collaborative filtering model based on autoencoders, utilized for personalized

**Table 4  Partition of datasets.**

| | ID | Ratio of training data | # Samples in trainging set | # Samples in testing set |
|---|---|---|---|---|
| Response time | RT_05 | 5% | 93,692 | 1,780,146 |
| | RT_10 | 10% | 187,384 | 1,686,454 |
| | RT_15 | 15% | 281,076 | 1,592,762 |
| | RT_20 | 20% | 374,768 | 1,499,070 |
| Throughput | TP_05 | 5% | 91,563 | 1,739,690 |
| | TP_10 | 10% | 183,125 | 1,648,128 |
| | TP_15 | 15% | 274,689 | 1,556,564 |
| | TP_20 | 20% | 366,251 | 1,465,002 |

recommendation tasks in recommendation systems. It employs autoencoders to learn the hidden structure within user-item rating matrices, thereby facilitating the learning and representation of user interests and item features.

Deep Learning-based Communication Adaptive Network (DLCANet) (*Kim et al., 2024*): DLCANet is a deep learning-based model specifically designed for QoS prediction in robotic communication networks. By integrating CNN, LSTM, GNN, and attention mechanisms, DLCANet captures spatial, temporal, and topological features of communication data, making it highly effective for dynamic and complex network environments. Through meticulous data preprocessing and a custom loss function that emphasizes critical QoS metrics like latency and bandwidth.

## Results and discussion

In this work, we evaluated the performance of seven algorithms—RSNMF, NAMF, GeoMF, DCALF, AutoRec, DLCANet, and RA-QoS—on two benchmark QoS datasets: RT and TP. The experiments were conducted across varying training data ratios, and the evaluation metrics used were MAE and RMSE. The detailed results are summarized in Tables 5 and 6.

As introduced in "The Proposed Method: Ra-qos Model", the RA-QoS framework supports two input modes: I-RA-QoS (item-based) and U-RA-QoS (user-based), each offering different perspectives of QoS matrix completion. To determine the most effective variant for practical deployment, we conduct a comprehensive comparison of both I-RA-QoS and U-RA-QoS across multiple datasets and under various combinations of Preprocessing Bias (PB) and Training Bias (TB). Experimental results consistently demonstrate that I-RA-QoS achieves better prediction accuracy, stronger robustness to sparsity, and faster convergence than U-RA-QoS. Therefore, we adopt I-RA-QoS with PB.2 and TB.2 as the default configuration of the final RA-QoS model, ensuring high accuracy and effective debiasing while preserving architectural simplicity.

Table 5 shows the MAE results, where RA-QoS consistently demonstrates strong performance, often achieving the lowest MAE values. For the RT dataset, RA-QoS outperforms all models in most cases, with the lowest MAE of 0.3835 at RT_20, demonstrating its ability to minimize errors. In the TP dataset, RA-QoS remains

**Table 5  Experimental results on metric of MAE.**

| Datasets | Metric | RSNMF | NAMF | GeoMF | DCALF | AutoRec | DLCANet | RA-QoS |
|---|---|---|---|---|---|---|---|---|
| RT_05 | MAE | 0.5438 | 0.5465 | 0.5305 | 0.5127 | 0.5467 | 0.5341 | 0.5126 |
| RT_10 | MAE | 0.4868 | 0.4976 | 0.4827 | 0.4544 | 0.5055 | 0.4805 | 0.4355 |
| RT_15 | MAE | 0.4492 | 0.4625 | 0.4495 | 0.4346 | 0.4598 | 0.4519 | 0.4061 |
| RT_20 | MAE | 0.4371 | 0.4360 | 0.4366 | 0.4246 | 0.4482 | 0.4291 | 0.3835 |
| TP_05 | MAE | 21.4302 | 20.2101 | 24.7465 | 18.6237 | 21.3118 | 20.3118 | 19.4392 |
| TP_10 | MAE | 17.2305 | 17.0126 | 22.4728 | 15.3430 | 17.031 | 19.0131 | 16.2878 |
| TP_15 | MAE | 14.6880 | 15.6547 | 17.7908 | 14.0664 | 15.0156 | 15.7456 | 15.1795 |
| TP_20 | MAE | 14.3654 | 14.6482 | 16.2852 | 13.5491 | 14.2265 | 15.4263 | 14.5654 |
|  | Win/Loss | 21/27 | 17/31 | 12/36 | 44/4 | 15/33 | 19/29 | 40/8 |
|  | Friedman rank | 5.000 | 5.571 | 6.286 | 1.714 | 5.857 | 5.286 | 2.286 |

**Table 6  Experimental results on metric of RMSE.**

| Datasets | Metric | RSNMF | NAMF | GeoMF | DCALF | AutoRec | DLCANet | RA-QoS |
|---|---|---|---|---|---|---|---|---|
| RT_05 | RMSE | 1.4032 | 1.3995 | 1.3152 | 1.3731 | 1.3730 | 1.3975 | 1.3704 |
| RT_10 | RMSE | 1.2689 | 1.2694 | 1.2191 | 1.2450 | 1.2678 | 1.2718 | 1.2601 |
| RT_15 | RMSE | 1.2067 | 1.2178 | 1.1742 | 1.2001 | 1.1923 | 1.1967 | 1.2040 |
| RT_20 | RMSE | 1.1588 | 1.1592 | 1.1528 | 1.1759 | 1.1681 | 1.1691 | 1.1573 |
| TP_05 | RMSE | 60.7994 | 53.9572 | 57.7842 | 51.4123 | 55.5352 | 54.7339 | 53.5613 |
| TP_10 | RMSE | 50.5298 | 46.0215 | 49.2456 | 45.9013 | 48.4771 | 50.2944 | 47.3237 |
| TP_15 | RMSE | 45.2647 | 43.6522 | 45.3255 | 42.6235 | 44.5246 | 45.8946 | 44.3028 |
| TP_20 | RMSE | 43.5882 | 42.3523 | 43.9845 | 41.2194 | 43.0654 | 43.7362 | 42.5971 |
|  | Win/Loss | 11/37 | 24/24 | 28/20 | 35/13 | 25/23 | 12/36 | 33/15 |
|  | Friedman rank | 6.429 | 4.571 | 4.000 | 3.000 | 4.429 | 6.286 | 3.286 |

competitive, outperforming the other models except in a few instances, such as at TP_20 where DCALF slightly surpasses it (MAE = 13.5491 for DCALF *vs.* 14.5654 for RA-QoS). The Win/Loss analysis highlights RA-QoS's effectiveness, achieving 40 wins against only eight losses, the second-best performance after DCALF (44 wins, four losses).

The RMSE results in Table 6 further validate the superiority of RA-QoS in both datasets. For RT, RA-QoS achieves an RMSE of 1.1573 at RT_20, which is marginally higher than GeoMF's RMSE of 1.1528 but still ranks among the best-performing models. In the TP dataset, RA-QoS consistently delivers competitive results, achieving RMSE values close to DCALF, which leads in several scenarios (*e.g.*, TP_20, RMSE = 41.2194 for DCALF *vs.* 42.5971 for RA-QoS). The overall win/loss analysis shows RA-QoS securing 33 wins and 15 losses, confirming its robustness compared to most other models.

The Friedman Rank analysis, presented in Tables 5 and 6, offers a comprehensive view of the models' performance. RA-QoS ranks second for MAE (rank = 2.286) and third for RMSE (rank = 3.286), reflecting its consistent accuracy and reliability. In contrast, models like RSNMF and NAMF generally exhibit weaker performance, with ranks of 5.000 and 5.571 for MAE and 6.429 and 4.571 for RMSE, respectively.

The results highlight RA-QoS's ability to achieve close-to-optimal predictions across varying data sparsity levels. While DCALF occasionally outperforms RA-QoS, especially in low-sparsity conditions, RA-QoS demonstrates competitive performance with minimal differences in accuracy. For instance, at TP_20, the MAE and RMSE values for RA-QoS are within a narrow margin of DCALF's results, showcasing RA-QoS's robustness.

Additionally, RA-QoS strikes a balance between accuracy and versatility, performing well across both RT and TP datasets, regardless of the training data ratio. This versatility makes it a strong candidate for real-world QoS prediction tasks, where data sparsity and distribution vary significantly. While DCALF emerges as the top-performing model in specific scenarios, RA-QoS delivers highly competitive results with a consistent edge over most other models. Its strong performance across different metrics, combined with its adaptability, positions RA-QoS as a reliable and effective model for QoS prediction tasks.

To provide a more intuitive comparison, Fig. 4 contrasts the experimental results of the AutoRec model and the proposed RA-QoS model. It can be seen that RA-QoS consistently exhibits lower RMSE and MAE compared to AutoRec across both datasets, indicating its superior predictive performance. This highlights the advantage of the RA-QoS approach in improving the prediction of web service quality, making it a valuable enhancement over the original AutoRec methodology.

Conclusion. Overall, the experimental results on both RT and TP datasets validate the effectiveness of RA-QoS. It consistently achieves lower MAE and RMSE than baseline models under various data sparsity settings. Although DCALF slightly surpasses RA-QoS in a few individual cases, RA-QoS offers more stable performance overall, especially in highly sparse settings. This demonstrates its superior generalization and robustness, largely attributed to the integration of bias correction and hybrid loss design.

## Parameter sensitivity and convergence

In the experiment, achieving better results, _i.e._, lower RMSE and MAE, was not merely a matter of using the model directly. Instead, it involved numerous rounds of hyperparameter tuning, model adjustments, and comparisons with other factors affecting the prediction results. Therefore, this chapter will delve into the impact of various factors related to deep neural networks, including the network's depth, the number of neurons in each layer, and certain model hyperparameters, on the experimental outcomes. Additionally, we will conduct an ablation study to validate the positive effects of the improvements made to the model. The ablation study encompasses two aspects: one is the combination of PB and TB, and the other is the fusion of L1 and L2 norms as part of the loss function.

**Number of hidden layers**. Directly using the RA-QoS model to predict the two datasets yielded unsatisfactory results. The obtained RMSE and MAE values were considerably larger. Upon observing and analyzing the datasets used in the experiment, it was found that they exhibited significant variance and high data sparsity. Consequently, this article proposes increasing the number of hidden layers in the model, specifically adding layers to both the encoder and decoder, to enhance the model's learning capacity and enable it to handle such datasets more effectively. The comparison of experimental results under

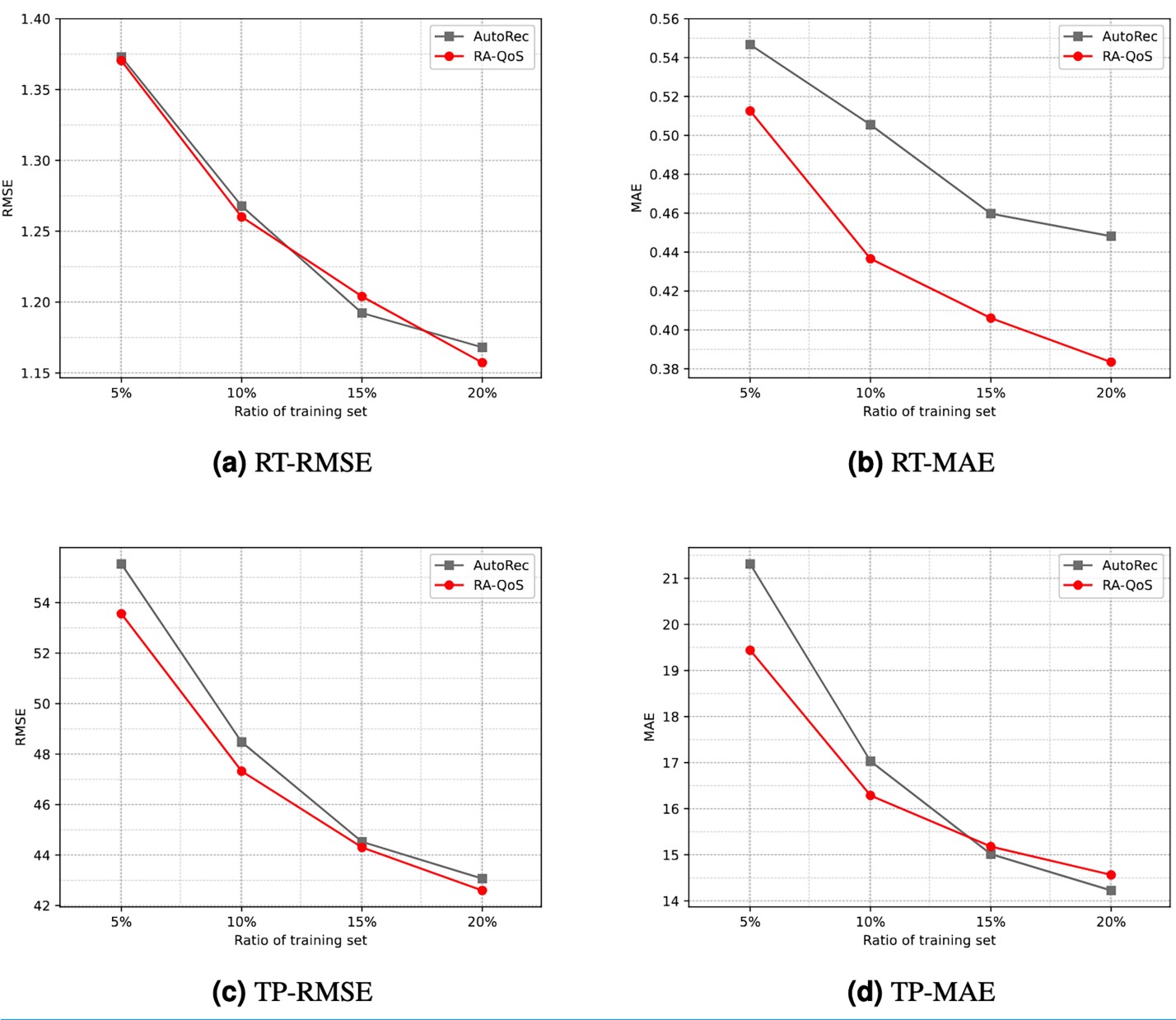

**Figure 4  Comparison of results between AutoRec and RA-QoS.** (A) RT-RMSE. (B) RT-MAE. (C) TP-RMSE. (D) TP-MAE.

different numbers of hidden layers is illustrated in Fig. 5. It can be observed that the experimental results are optimal when the number of hidden layers is three, *i.e.*, when an additional layer is added to both the encoder and decoder. However, as the number of hidden layers increases, both the efficiency and generalization ability of the model decline. This is because an excessive number of hidden layers increases the risk of overfitting, potentially leading to the model learning noise and details in the data excessively, thereby reducing the prediction accuracy of the model. Interestingly, for the dataset related to

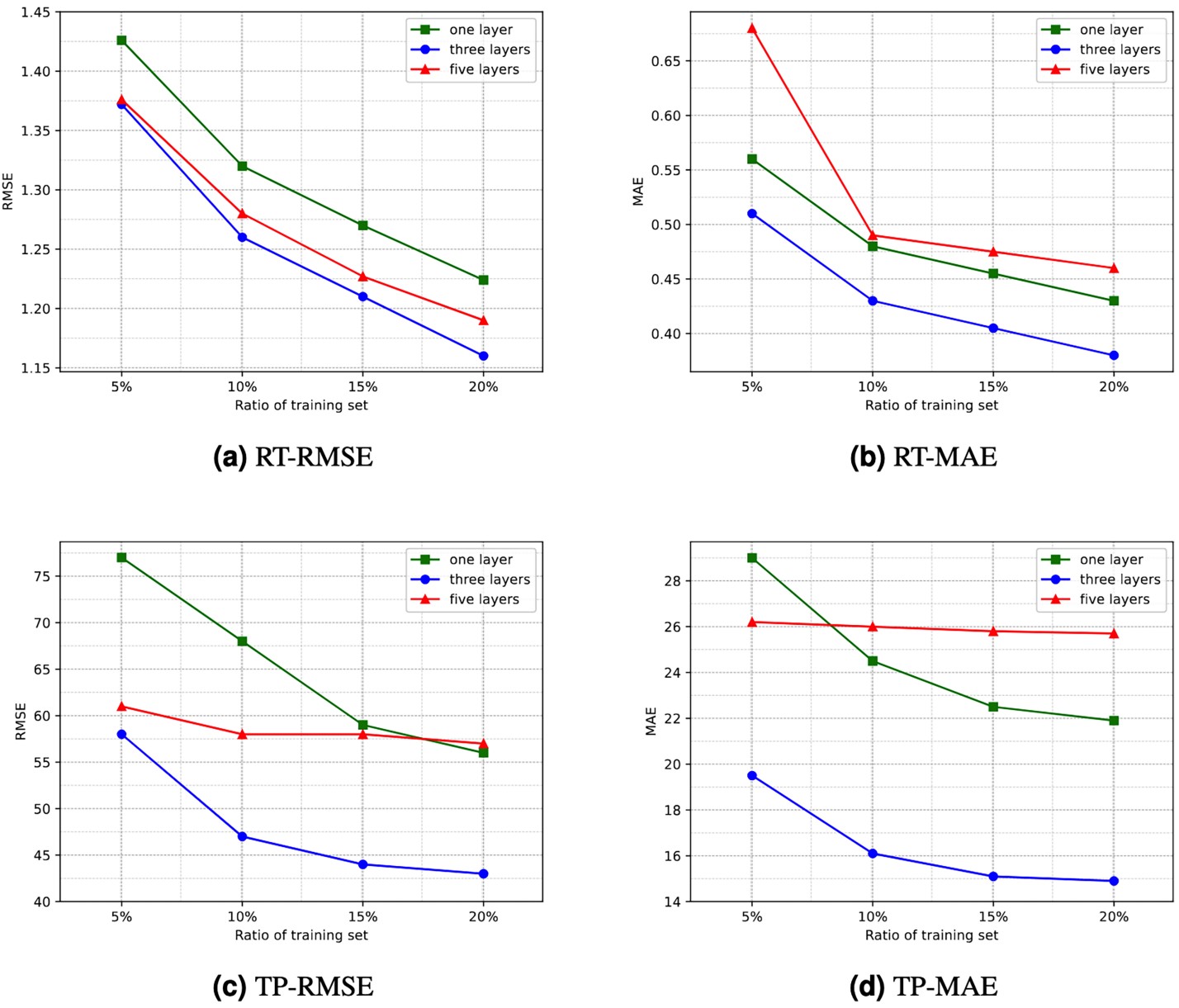

**(a)** RT-RMSE

**(b)** RT-MAE

**(c)** TP-RMSE

**(d)** TP-MAE

**Figure 5 Comparison of experimental results with respect to number of hidden layers.** (A) RT-RMSE. (B) RT-MAE. (C) TP-RMSE. (D) TP-MAE.

response time, directly using three hidden layers did not yield satisfactory results. Instead, achieving better results was observed by reducing one layer in the decoder. Therefore, in subsequent comparative experiments and ablation studies, a three-layer hidden structure was used, with slight variations in the decoder stage.

**Number of iterations**. The number of iterations (or training epochs) in deep neural networks has a significant impact on prediction results. With more iterations, the degree of fitting to the training data may increase, but this also raises the risk of overfitting and

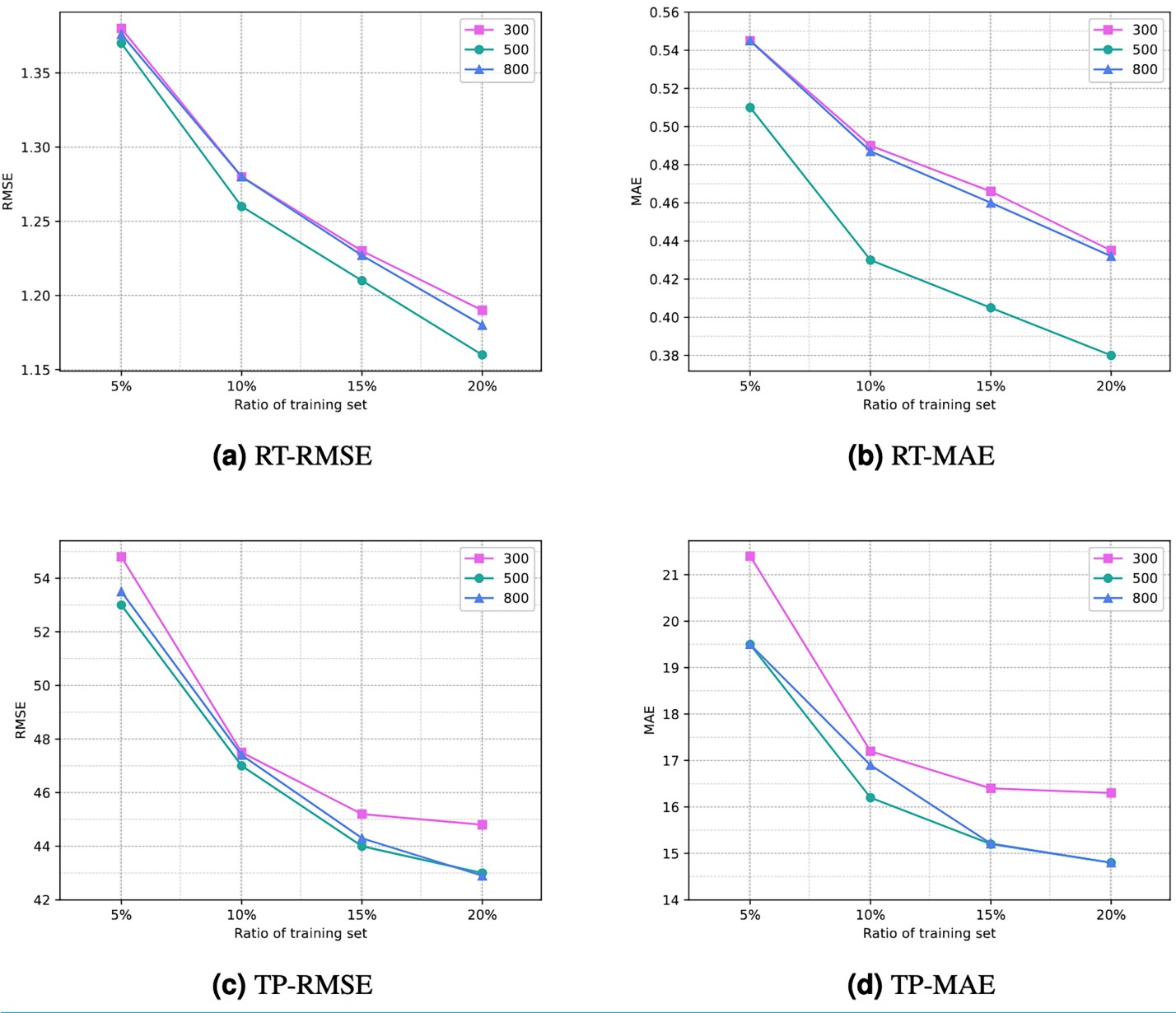

**Figure 6 Comparison of experimental results with respect to number of iterations.** (A) RT-RMSE. (B) RT-MAE. (C) TP-RMSE. (D) TP-MAE.

requires more computational time. Selecting the appropriate number of iterations involves balancing the trade-off between overfitting and underfitting, while also aiding in improving model generalization. Figure 6 illustrates the comparison of the impact of different numbers of iterations on experimental results. It can be noted that as the number of iterations increases, the results become increasingly favorable. However, after 500 iterations, the reduction in RMSE and MAE diminishes, and there may even be negative growth. Therefore, setting the number of iterations to 500 can yield a satisfactory prediction performance for the model.

**Table 7 The table of four scenarios in ablation study.**

|    | PBTB combination | Loss function upgraded |
| --- | --- | --- |
| A1 | × | × |
| A2 | √ | × |
| A3 | × | √ |
| A4 | √ | √ |

## Ablation study

As this article described, the experiment in this study involved the RA-QoS model, primarily manifested in two aspects: the incorporation of different combinations of PB and TB, and the modification of the loss function. However, it is necessary to verify whether these modifications have improved the accuracy of the prediction results. Therefore, this article conducts an ablation study using the method of controlling variables.

This study includes four scenarios: not including PB and TB combinations without modifying the loss function; including PB and TB combinations without modifying the loss function; not including PB and TB combinations while modifying the loss function; and including different combinations of PB and TB while modifying the loss function. For the convenience of visualization, we represent these four scenarios as A1, A2, A3, and A4, respectively. The details of configuration of A1, A2, A3, and A4 are shown in the Table 7.

The specific comparison results are illustrated in Fig. 7. It can be seen that the line corresponding to A4 is located at the bottom of the graph overall. This indicates that the prediction performance of scenario A4 is the best among the four scenarios, implying that the proposed model's improvement is effective.

## Robustness test

To evaluate the robustness of the RA-QoS model when facing noise, we designed a series of experiments that introduce different types and intensities of noise into the QoS data, and observe the model's performance under these noisy environments. The main goal of the experiment is to verify whether the RA-QoS model can maintain prediction accuracy stably in the presence of noise, and to compare its performance with baseline methods.

### Noise data injection methods

In the experiments, we introduced noise into the QoS data using the following methods:

**Gaussian noise**. Gaussian noise was added to each data point by sampling from a standard normal distribution. Specifically, each response time $r_{u,s}$ was modified as follows:

$$r'_{u,s} = r_{u,s} + \varepsilon_{u,s}$$

where $\varepsilon_{u,s} \sim \mathcal{N}(0, \sigma^2)$, and the noise standard deviation $\sigma$ was set to different values (0, 0.2, 1.0) to simulate different levels of noise intensity.

### Experimental procedure

**Dataset selection:** We used the WS-Dream dataset containing multiple services and user QoS metrics, including response time and throughput. The RA-QoS model was trained on

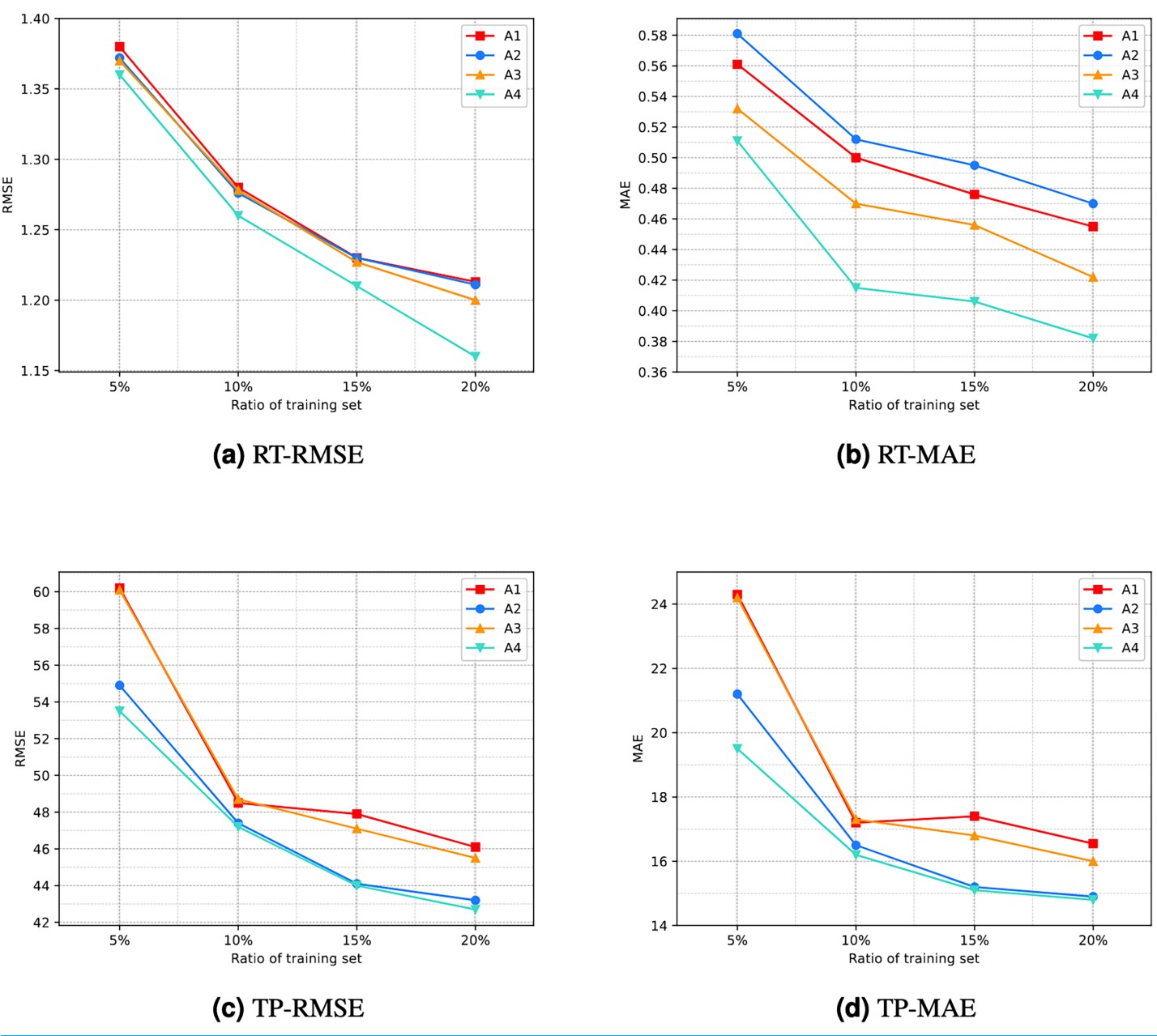

**Figure 7  Result of ablation experiment.** (A) RT-RMSE. (B) RT-MAE. (C) TP-RMSE. (D) TP-MAE.

both the original and noisy data, and evaluated using metrics such as MAE and RMSE. To evaluate the robustness of the RA-QoS model, we compared it with several baseline models that used in the former mentioned experiments.

We evaluated the robustness of the RA-QoS model under various noise levels. The RMSE values for different methods under different noise levels are shown in the table and the corresponding trend is illustrated in Fig. 8.

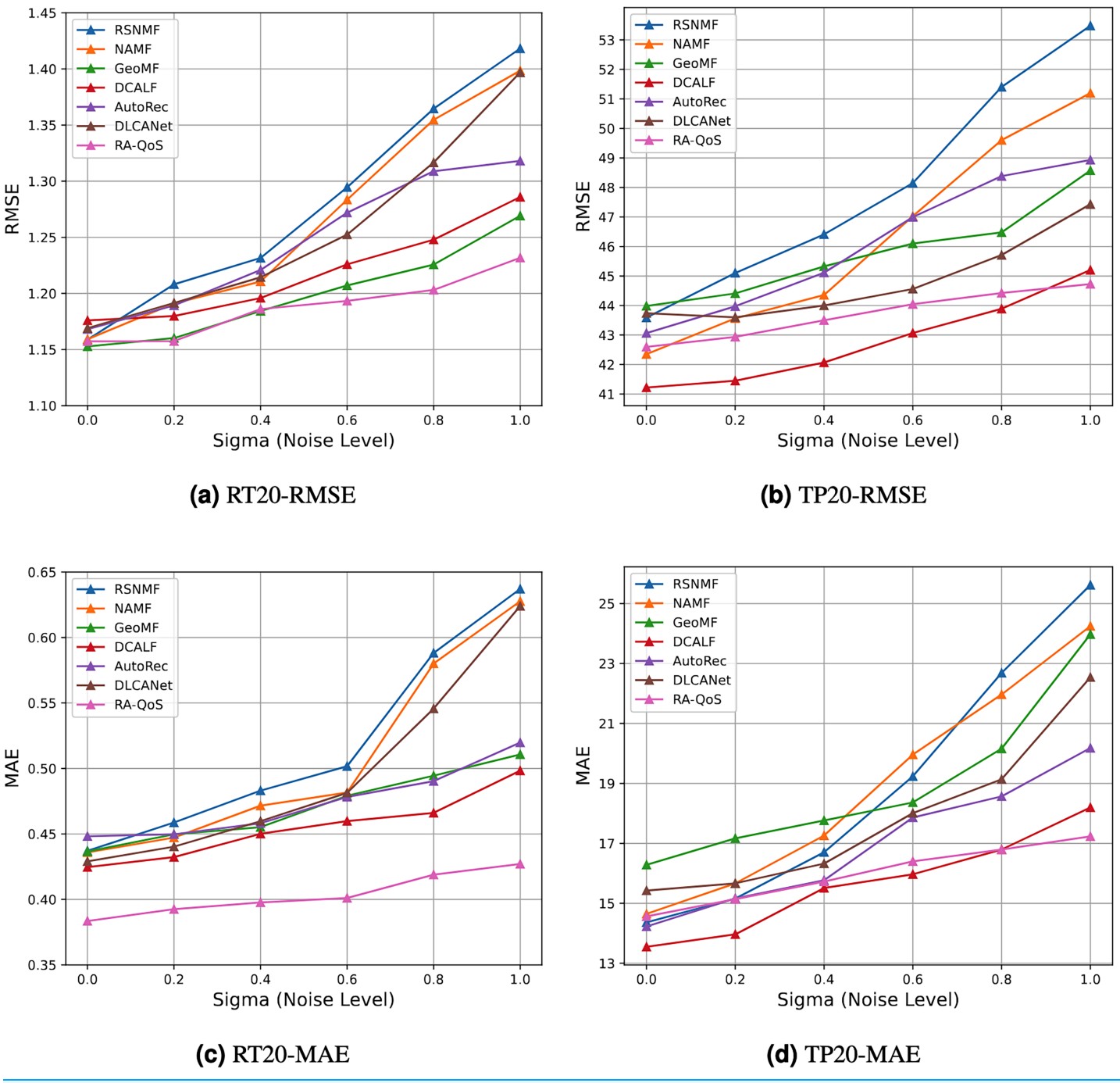

**Figure 8** **Result of robustness test.** (A) RT20-RMSE. (B) TP20-RMSE. (C) RT20-MAE. (D) TP20-MAE.

The results showed that as the noise intensity increased (*e.g.*, from $\sigma = 0$ to $\sigma = 1.0$), the MAE and RMSE for the RA-QoS model remained relatively stable, with only a slight increase in error. Even under high noise levels (*e.g.*, $\sigma = 1.0$), the RA-QoS model maintained low error rates, demonstrating strong robustness.

In contrast, the baseline methods, such as matrix factorization based RSNMF, NAMF and Collaborative Filtering based AutoRec, showed more unstable performance under high noise conditions. Specifically, with the introduction of missing data and high Gaussian noise, their MAE and RMSE values increased significantly, and their performance degraded faster. This indicates that the RA-QoS model is more stable when facing noise compared to these baseline methods. Through experiments with varying noise levels, we observed that the RA-QoS model remains stable and maintains low error rates even in the presence of high noise and missing data. In comparison, the baseline methods exhibited significant performance degradation under noisy conditions. This demonstrates that the RA-QoS model is robust and performs stably when confronted with data noise, making it a reliable choice for QoS prediction in real-world applications with imperfect data.

As the noise intensity (*i.e.*, the value of $\sigma$) increases, the RMSE values for all baseline methods increase, indicating a deterioration in performance as noise levels rise. Among the baseline methods, including RSNMF, NAMF, GeoMF, DCALF, AutoRec and DLCANet, RA-QoS exhibits the smallest increase in RMSE, demonstrating its superior robustness against noise. This result highlights the effectiveness of the RA-QoS model in handling noisy data, making it a more stable and reliable choice for real-world QoS prediction tasks.

RA-QoS consistently demonstrates the smallest increase in both RMSE and MAE as the noise level increases, across both datasets (RT_20 and TP_20). This indicates that RA-QoS is more robust to noise than other baseline methods. The model's ability to maintain low error rates, even under noisy conditions, makes it a reliable choice for QoS prediction in real-world environments.

Methods like RSNMF, NAMF, and AutoRec show a higher sensitivity to noise, with their error metrics increasing more significantly as the noise level rises. These methods struggle to handle the noise and exhibit performance degradation that RA-QoS avoids.

The GeoMF and DCALF models perform reasonably well but still show more sensitivity to noise compared to RA-QoS. While these models demonstrate better robustness than RSNMF and NAMF, they still experience a noticeable increase in RMSE and MAE values as noise levels grow.

Conclusion. This section reveals that the performance of RA-QoS is highly sensitive to bias configuration and network structure. The integration of PB.2 and TB.2 significantly enhances prediction accuracy and training efficiency. Moreover, the L1-L2 hybrid loss contributes to stronger robustness against noisy and outlier data. These findings confirm that the design choices in RA-QoS are effective in addressing challenges such as data sparsity, outliers, and convergence speed in deep QoS prediction.

## CONCLUSION AND FUTURE WORKS

For the task of QoS prediction, issues arise from large data volume and sparsity. This article addresses issues such as non-linear feature learning of data, and the inability to simultaneously consider local and global features, by proposing an enhancement of the AutoRec model used in recommendation systems, termed RA-QoS.

**Key findings**: By combining PB and TB and utilizing L1 and L2 norms, the issues caused by outliers and various biases are resolved. Experimental results demonstrate that, with appropriate hyperparameter selection, the proposed model can achieve relatively high accuracy in practical QoS prediction tasks. This article discusses the selection of relevant hyperparameters. It is noteworthy that we also conducted ablation experiments to demonstrate the effectiveness of the model.

**Limitation**: Despite its effectiveness, the RA-QoS model's performance depends on hyperparameter tuning. Additionally, the current model structure may benefit from further refinement of the encoder-decoder architecture, as well as from integrating search algorithms for more efficient hyperparameter optimization.

**Future work**: In the future, it is possible to design more optimal encoder and decoder structures, such as replacing the decoder with Logistic Regression, thereby transforming the recommendation problem into a classification problem. Additionally, in terms of hyperparameter tuning, employing search algorithms could enhance the efficiency of optimization. The proposed RA-QoS model offers a promising and practical approach for accurate, robust QoS prediction, with future improvements focused on optimizing encoder-decoder structures and hyperparameter tuning strategies.

### Funding

This work is supported by the Science and Technology Research Program of Chongqing Municipal Education Commission (Grant Nos. KJQN202303203, KJZD-M202203201), Natural Science Foundation of Chongqing, China CSTC (Grant No. CSTB2023NSCQ-MSX0981), Doctoral Fund of Chongqing Industry Polytechnic College (No. 2023GZYBSZK3-03). There was no additional external funding received for this study. The funders had no role in study design, data collection and analysis, decision to publish, or preparation of the manuscript.

### Grant Disclosures

The following grant information was disclosed by the authors:
Science and Technology Research Program of Chongqing Municipal Education Commission: KJQN202303203, KJZD-M202203201.
Natural Science Foundation of Chongqing, China CSTC: CSTB2023NSCQ-MSX0981.
Chongqing Industry Polytechnic College: 2023GZYBSZK3-03.

### Competing Interests

The authors declare that they have no competing interests.

### Author Contributions

- Shun Fu conceived and designed the experiments, performed the experiments, analyzed the data, performed the computation work, prepared figures and/or tables, authored or reviewed drafts of the article, and approved the final draft.
- Junnan Li analyzed the data, prepared figures and/or tables, and approved the final draft.

- Lufeng Wang analyzed the data, performed the computation work, prepared figures and/or tables, and approved the final draft.

## Data Availability

The WS-Dream dataset is available at https://wsdream.github.io.

## Supplemental Information

Supplemental information for this article can be found online at http://dx.doi.org/10.7717/peerj-cs.2928#supplemental-information.

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
