# Peer review of "RA-QoS: a robust autoencoder-based QoS predictor for highly accurate web service QoS prediction"

_PeerJ Computer Science, doi:10.7717/peerj-cs.2928_

## Round 0.1 · original submission · Major Revisions

Based on the overall comments by reviewers, the authors are provided with the opportunities to revise major points raised, particularly on the results and findings as well as justifications or explanations of the findings. Please see the review comments in more details.

Reviewer 1 ·

Basic reporting

- The abstract should include the objectives, innovation and quantified results (generally). It should fall within the range of 150 to 250 words.
- Represent summary of results, limitations, and conclusions at the end
of abstract.
- The introduction should include context, formal problem statement, gap of study, objectives, contributions (in bullet points) and Paper Organization.; these elements should be explained in one paragraph.
-- The formal definition of the problem should be written in the introduction.
-- The main novelties of the study should be explained in the introduction.
- Review the related state of the art methods in a separate section as "related works (Section 2)". In this section the merits and demerits of the related methods should be explained in a table.
- The reasons behind the selection of autoencoder as the quality of service predictor in the suggested method should be explained in the manuscript.
- The quality criteria for the services should be explained in the manuscript?
- How the quality was calculated in the study?
- The input and output of the developed autoencoder should be explained?
- How the autoencoder classifies the services based on their quality?

- Which drawbacks of the related state-of-the-art works have been addressed by the proposed method.
- A flowchart of the method should be inserted in the first part of proposed method section and the subsections of this section should be written based on the flowchart.
- Its hard to justify on the method because the proposed method should be explained in detailed and in step by step form regarding the flowchart.
- In the experiment system section, the platform, implemented codes, evaluation criteria, research questions and datasets should be explained in different subsections.
- The obtained results are not enough to justify the proposed method.
- the paper lacks a detailed comparison of the proposed method with other existing methods. The authors are suggested to compare the results with the recently published methods in terms of different related criteria.
- The results and discussion should be moved to a separate subsection as "Results and discussion". The discussion should be made based on the research questions.
- There is not enough explanation and discussion for the obtained results and figures of your experiments.
-statistical analyses, controls, sampling mechanism, and statistical reporting (e.g., P-values, CIs, effect sizes) should be described in the manuscript.
- The findings and limitations of the study were not clearly stated in the conclusion.

Experimental design

In the experiment system section, the platform, implemented codes, evaluation criteria, research questions and datasets should be explained in different subsections.
- The obtained results are not enough to justify the proposed method.

Validity of the findings

The results and discussion should be moved to a separate subsection as "Results and discussion". The discussion should be made based on the research questions.
- There is not enough explanation and discussion for the obtained results and figures of your experiments.
-statistical analyses, controls, sampling mechanism, and statistical reporting (e.g., P-values, CIs, effect sizes) should be described in the manuscript.

Reviewer 2 ·

Basic reporting

The paper addresses the Web service QoS prediction problem and proposes to use deep neural networks to design a robust QoS prediction method. The paper designs an autoencoder-based RA-QoS model for the QoS prediction problem, and attempts to model the QoS prediction problem more accurately and give a reasonable prediction solution. However, the paper has the following three major issues: (1) the paper does not compare to and analyze the state-of-the-art deep learning-based QoS prediction algorithms. (2) The author states in the title of the paper that the proposed prediction method is robust, but the method description and experiments in the main body do not reflect the robustness of the proposed algorithm. (3) The paper has more problems in presentation, which leads to poor readability of the paper.

Experimental design

1. The paper uses the Deep neural networks (DNN) based method, but in the comparison experiments, it has not been compared with other DNN-based methods. Therefore, it is recommended that the authors add DNN-based methods for comparison, such as which methods are mentioned in lines 111-112 in Related Work section. Moreover, the methods currently used for comparison, RSNMF and AutoRec, were proposed relatively early, and it is recommended to select some recent relevant studies to increase the solidness of the paper’s conclusions.
2. In the title and abstract, the paper emphasizes that the proposed QoS prediction algorithm is robust, but it does not reflect this point in sections of model construction and experimental analysis, so it is suggested that experiments should be conducted to confirm this point.

Validity of the findings

1. The experimental results of the compared method DCALF on Response Time dataset performs the same as that of the original paper (Wu et al., 2022), but there is a big difference on Throughput dataset, what is the reason for this difference? The authors should give an in-depth elaboration and clarification.
2. Tables 1 and 2 mention 8 all possible combinations of PB consisting of z_1,z_2 and z_3 and 4 all possible combinations of TB consisting of z_4 and z_5 for calculating Ψ and ϕ, respectively, but there is no mention in the experimental part of how the parameters z_1,z_2,z_3,z_4 and z_5 take their values and how they affect the algorithm performance.

Additional comments

1. When the paper expresses some data or information, it is recommended to use a more intuitive way of expression in order to facilitate the reader’s reading and understanding. For example, the first column “Datasets” in Table 4 can be distinguished directly by dataset name and densities instead of custom IDs.
2.The paper has some inconsistencies in presentation or misdescriptions that need to be further verified. For example:
(1) In line 114, the text is expressed as ‘In 2021,...(Zhu et al., 2018)’, Zhu et al. published their academic work in 2018, and its year does not correspond.
(2) Equation (4) is used to calculate P_Q, which is meant to calculate Pa according to the text (lines 149-153), whether it is written incorrectly.
(3) Line 191, x×O(|M|)+|M|×x×O(|M|) should be x×O(|M|)+O(|M|)×x×O(|M|).
3. There are still more problems in the grammar and format of the paper writing, and further checking is recommended.

Reviewer 3 ·

Basic reporting

The english is fine, but could use basic improvements. There is an example of an acronym being used that was not introduced prior.

Experimental design

The experimental design is good.

Validity of the findings

The validity seems fine, but I have one concern due to the citations. The authors are comparing their results to works that are not recent enough. This would be okay because it is possible there are not many recent works in the field.

Additional comments

For someone not an expert in the specific parameters in your field, it was a bit difficult interpreting some of the results. Firstly, with regards to Figure 6, it should be before the the conclusion. More importantly however, is the y-axis scaling. I see sometimes (a) the RMSE y-axis is from 1.15 to 1.35, but then in (b) the y-axis is 42 to 54. Without more context I do not know if an RMSE of 42 to 54 is good or bad?

Also, the x-axis being ratio of training set is interesting. It looks like you are showing that as the ratio of the training set increases from 5% to 20% the RMSE decreases. This concept is not too clear however. Are you increasing the quantity of the training data itself?

---

## Round 0.2 · Minor Revisions

Thank you for improving your manuscript after the first round of review. Please address the remaining issues identified by Reviewer 2.

Reviewer 2 ·

Basic reporting

The quality of the paper has been significantly improved after the last round of revisions, but there are still some issues that need further refinement:
(1) The recently added algorithms in the experiments (e.g., DLCANet) should be included in Table 1.
(2) The schematic diagram of the RA-QoS model's architecture (Figure 3) is not very clear in conveying its intended meaning, and it is recommended to redraw it.
(3) In Section 4 (particularly Sections 4.4 and 4.5), the explanation of how the I-RA-QoS model and U-RA-QoS are combined to form the final RA-QoS model is not clearly stated.

Experimental design

The authors are required to publicly release the algorithm implementations and datasets involved in the experiments of the paper, so that other researchers can replicate the results and conclusions in the paper.

Validity of the findings

To facilitate the readers' understanding, it is necessary to clearly present the experimental conclusions at the end of each experiment in Sections 5.6 and 5.7.

---

## Round 0.3 · accepted · Accept

Thanks to the authors for their efforts to improve the article. It can be accepted currently. Congrats!

Reviewer 2 ·

Basic reporting

No comment.

Experimental design

No comment.

Validity of the findings

No comment.